# Exploring the Interplay between Metabolic Pathways and Taxane Production in Elicited *Taxus baccata* Cell Suspensions

**DOI:** 10.3390/plants12142696

**Published:** 2023-07-19

**Authors:** Edgar Perez-Matas, Pascual Garcia-Perez, Begoña Miras-Moreno, Luigi Lucini, Mercedes Bonfill, Javier Palazon, Diego Hidalgo-Martinez

**Affiliations:** 1Department of Biology, Healthcare and the Environment, Faculty of Pharmacy and Food Sciences, University of Barcelona, 08028 Barcelona, Spain; epm093@hotmail.com (E.P.-M.); mbonfill@ub.edu (M.B.); 2Department for Sustainable Food Process, Università Cattolica del Sacro Cuore, Via Emilia Parmense 84, 29122 Piacenza, Italy; pasgarcia@uvigo.es (P.G.-P.); mariabegona.mirasmoreno@unicatt.it (B.M.-M.); luigi.lucini@unicatt.it (L.L.); 3Nutrition and Bromatology Group, Department of Analytical and Food Chemistry, Faculty of Food Science and Technology, Ourense Campus, Universidade de Vigo, 32004 Ourense, Spain

**Keywords:** metabolomics, biomarkers, paclitaxel, taxanes, coronatine, salicylic acid

## Abstract

*Taxus* cell cultures are a reliable biotechnological source of the anticancer drug paclitaxel. However, the interplay between taxane production and other metabolic pathways during elicitation remains poorly understood. In this study, we combined untargeted metabolomics and elicited *Taxus baccata* cell cultures to investigate variations in taxane-associated metabolism under the influence of 1 µM coronatine (COR) and 150 µM salicylic acid (SA). Our results demonstrated pleiotropic effects induced by both COR and SA elicitors, leading to differential changes in cell growth, taxane content, and secondary metabolism. Metabolite annotation revealed significant effects on N-containing compounds, phenylpropanoids, and terpenoids. Multivariate analysis showed that the metabolomic profiles of control and COR-treated samples are closer to each other than to SA-elicited samples at different time points (8, 16, and 24 days). The highest level of paclitaxel content was detected on day 8 under SA elicitation, exhibiting a negative correlation with the biomarkers kauralexin A2 and taxusin. Our study provides valuable insights into the intricate metabolic changes associated with paclitaxel production, aiding its potential optimization through untargeted metabolomics and an evaluation of COR/SA elicitor effects.

## 1. Introduction

Plants belonging to the genus *Taxus*, coniferous trees or shrubs known vernacularly as yews, constitute major sources of various bioactive compounds, such as paclitaxel (PTX) and its derivatives. PTX, also known by its commercial name Taxol^®^, is a tricyclic diterpenoid widely used in the treatment of various types of cancer, including ovarian, breast, and squamous cancers [1]. This potent antitumor compound was first discovered decades ago in the bark of the Pacific yew tree (*Taxus brevifolia*) [2] and its pharmacological activity is mainly due to the presence of an ester side chain at C-13, an A ring, a C-2 benzoyl group, and an oxetane ring in its structure [3]. Further studies also demonstrated that the presence of a *m*-azido substituent in the C-2 benzoyl ring contributes to the high biological activity of this compound; molecular modelling studies performed with the C-2 benzoyl ring of paclitaxel indicated that it fits into a pocket formed by His227 and Asp224 on the β-tubulin and that the 2-*m*-azido is capable of enhancing the interaction between the benzoyl group and the side chain of Asp224 [4]. In contrast with other microtubule-targeting anticancer agents, these features allow PTX to induce mitotic arrest by binding and stabilizing the dimers of β-tubulin, thus preventing microtubule dynamism and cell division [5].

Since its approval in 1992 by the Food and Drug Administration (FDA), the demand for PTX and its derivatives has not stopped growing [6]. However, PTX is a poorly soluble diterpene pseudoalkaloid with a very complex structure; its content in yew trees is extremely low, approximately 0.01% of ligneous dry weight, an amount that fluctuates considerably across the seasons [7]. According to Bedi et al. [8], a single course of treatment requires between 2.5 and 3 g of PTX, which implies a need for harvesting the inner bark of trees aged between 8 and 60 years. In this scenario of very low content, slow growing wild trees, complex synthetic routes, and high purification costs, alternative ways of obtaining PTX have been explored. PTX can also be obtained by semi-synthesis, starting from more abundant taxanes, such as 10-deacetylbaccatin III and baccatin III, which can be extracted from yew needles, a natural and renewable source [9]. Additionally, the semi-synthesized compound known as docetaxel, or Taxotere^®^, now widely applied in anti-cancer therapy, is more soluble in water than PTX and 2.7-fold more effective against certain cancers [10]. Nevertheless, the biotechnological production of PTX by means of optimized *Taxus* spp. cell cultures is the current method of choice as cell suspensions can be grown under controlled and cost-effective conditions [11,12,13,14,15]. Moreover, to increase taxane production, various strategies have been considered, such as the optimization of culture conditions, selection of high-producing cell lines, and use of elicitors and precursors [16,17].

Although the PTX biosynthetic pathway has been extensively elucidated, several hydroxylation steps are still unknown. Intriguingly, the pathway may not be linear but rather a network of anastomosing routes with common nodes. The resulting lack of clarity about the order of the reactions and the involvement of non-commercially available taxane substrates may have substantially increased the difficulty of identifying the unknown enzymes in the pathway [18].

Multi-omics represents a valuable high-throughput and comprehensive methodology that has been widely applied in the molecular study of plants [19]. Untargeted metabolomics is a useful strategy that provides a shortcut to systematically analyze and compare primary and specialized metabolisms, both under normal and elicited conditions [20]. In *Taxus* species, the first metabolomic analysis was carried out by Ketchum et al. [21] in 2003; they profiled the differential metabolites of *Taxus media* cell cultures with or without methyl jasmonate (MeJA) elicitation treatments. Since then, only a few more studies have involved a comparative metabolomics analysis of *Taxus* sp. For instance, a metabolomics approach with liquid chromatography hyphenated to an ion-trap time-of-flight mass spectrometer (LC-TOF-MS) was used to study the variations in taxoid biosynthesis in cultured seedlings of *Taxus mairei* [22]. Yu et al. [23] revealed metabolic variations between two endangered *Taxus* species from the Himalayas by means of comparative metabolomics; whereas, Hao et al. [24], through comparative proteomic analysis, revealed variations in the metabolism of paclitaxel and other metabolites between *T. media* and *T. mairei*. Finally, Zheng et al. [25] studied the response and defense mechanisms of *Taxus chinensis* leaves subjected to enhanced UV-A radiation using proteomics and metabolomics; Zhou et al. [26] used a metabolomics approach to investigate the variations in metabolites, including taxoids and flavonoids, between *T. mairei*, *T. cuspidata*, and *T. media*.

These previous studies have utilized multi-omics approaches to investigate *Taxus* species and the variations in their metabolite production. Additionally, many interesting data related to elicitation have been reported by Alcalde et al. [27]. However, only limited research has focused on understanding the metabolic changes induced by abiotic stressors, such as salicylic acid (SA) and coronatine (COR), in *Taxus baccata* cell suspensions. Therefore, in the present study, untargeted metabolomics was applied to explore the differential metabolic variations in *T. baccata* cell suspensions under abiotic stress caused by SA (150 µM) and COR (1 µM), with a particular emphasis on elucidating the influence of these elicitors on the relationship between primary and specialized metabolism. By identifying potential biomarkers associated with the elicitation of taxane production, we hoped to obtain new insights into the underlying mechanisms that influence taxane yields. Our aim was also to bridge existing knowledge gaps regarding taxane metabolism and, thus, contribute to the advancement of biotechnological systems for efficient and sustainable paclitaxel production.

## 2. Results

### 2.1. Growth and Viability of T. baccata Cell Cultures

Cell suspensions were first cultured in the corresponding optimum growth media for 12 days at an inoculum of 300 g/L and the biomass obtained was transferred to the optimum production media for 24 days to stimulate the production of taxanes to the detriment of growth. At the beginning of this stage, the inoculum was set again to 300 g/L; then, suspensions were elicited with COR 1 µM or SA 150 µM and samples were taken every 8 days.

Figure 1 depicts the growth profile and viability of cell suspensions under the various treatments. The biomass did not exhibit significant growth throughout the 24-day experiment under control conditions or SA treatment. Only the COR treatment induced a significant decrease in fresh biomass on day 24 (Figure 1a). Dry weight was significantly reduced by COR on day 8; but, the values subsequently recovered and remained constant until the end of the experiment (Figure 1b). Although there were minimal differences in significant growth, the viability of the cell cultures significantly decreased from the 8-h point until the end of the experiment under the influence of COR. In the case of SA treatment, a significant reduction in viability was only observed on day 16; whereas, the control showed a decrease in viability on day 24 (Figure 1c). The viability of COR-treated samples decreased by almost 25% from the beginning of the experiment (95%) to day 24 (72%). In contrast, cell viability in control and SA-treated samples ranged from 95% on day 0 to 87 and 81%, respectively.

### 2.2. Taxane Content

To determine the taxane content and the variations between the control and elicited conditions, a target quantification by HPLC was carried out on the following taxanes of interest: 10-deacetylbaccatin III (DABIII), baccatin III (BACIII), cephalomannine (CEPH), and paclitaxel (PTX).

As depicted in Figure 2a, total taxane contents greater than 7 mg/L were not detected in any of the experimental conditions. Despite this, elicitation significantly boosted taxane production in comparison to the control conditions. The highest total taxane contents were detected in SA-treated samples on day 8 of culturing, followed by COR-treated samples. At this time point, the contents reached 6.56 mg/L and 4.76 mg/L, respectively, which represented a 3.90-fold and 2.83-fold increase compared to the control. Subsequently, the taxane contents in the elicited samples decreased until the end of the experiment; although, they followed different patterns. In COR-treated samples there was a gradual decline until the minimum value was reached on day 24 of culturing (2.17 mg/L); whereas, in the SA-elicited samples, a sharp decrease was observed on day 16 (2.01-fold), reaching constant values (approx. 3 mg/L) thereafter. On the other hand, in control conditions, very low taxane levels were detected throughout the experiment. The results indicated that up to day 16, the total taxane content remained relatively constant (approx. 1.70 mg/L). However, similarly to the SA-elicited cell cultures, a sudden decrease in the taxane content occurred at the end of the experiment (0.72 mg/L).

Among the individual taxanes analyzed (DABIII, BACIII, CEPH and PTX), the elicitors induced a clear increase of PTX during the study period (Figure 2b). The highest amount of PTX was found on day 16 under SA elicitation (64.61% of the total); but, its contents in this treatment were highly variable, depending on the day of culture. Accordingly, on day 8 of post-elicitation, PTX reached a value of 2.40 mg/L, which corresponds to 59.75% of the total taxane content; whereas, at the end of the experiment, it had decreased to only 0.30 mg/L (9.23%). In contrast, the PTX content followed a more uniform trend in the samples elicited with COR, representing approx. 50% of the total taxane content throughout the experiment. In non-elicited conditions, PTX always accounted for less than 20% of the total taxanes. Regarding CEPH, the other taxane with a lateral chain, it is notable that the levels did not change over time in control conditions (0.46 mg/L). However, similarly to PTX, both elicitation treatments increased the CEPH contents in different ways. The highest CEPH levels were found at the beginning of the experiment under COR elicitation (1.60 mg/L), declining significantly thereafter to only 17.03% and 21.12% of the total taxanes on days 16 and 24, respectively. In contrast, in SA-treated samples, the CEPH contents increased progressively until reaching a maximum level (1.46 mg/L) on day 24 of culture, when it accounted for 44.92% of the total taxanes.

On the other hand, taxanes without a side chain, such as DABIII and BACIII, never reached values greater than 1 mg/L in any experimental conditions. Nevertheless, their production patterns and proportions with respect to the total taxane content varied, not only over time but also depending on the treatment. DABIII contents were the same in both control and COR-treated samples, tending to increase from day 8 (0.50 mg/L) to day 16 (0.8 mg/L), when they represented 47.56% and 22.63% of the total taxanes, respectively; subsequent values of DABIII were close to 0. In SA-elicited samples, the opposite trend was observed and the highest amount (0.8 mg/L) was reached on day 24 (24.49%). In the case of BACIII, the highest contents in the control and SA-treated cell cultures were achieved on day 8 of culturing (0.54 mg/L and 0.86 mg/L, respectively). Under COR elicitation, this value was not only lower but was also reached later, on day 24 (0.51 mg/L). Additionally, the proportion of BACIII was higher in control conditions (31.84% on day 8) than in either of the elicited cell cultures (23.56% in COR and 21.48% in SA on day 24).

Finally, the taxane contents were determined both inside the producer cells and in the culture medium. According to the results, most of the taxanes remained inside the cells, especially in the control and COR-treated samples, where less than 1 mg/L of the total contents were excreted into the culture medium. SA cultures exhibited a similar pattern of partitioning, except on day 8, when the extracellular content (1.81 mg/L) was 3.11-fold and 13.85-fold higher than in the control and COR cultures, respectively. Therefore, the elicitation treatments did not greatly stimulate the release of taxanes into the medium. Concerning the partitioning of individual taxanes, DABIII and BACIII were mostly released extracellularly whereas PTX, and especially CEPH, remained almost entirely inside the cells. In the case of PTX, it should be noted that on day 8 in SA-elicited cultures, approx. 27% of this taxane (1.076 mg/L) was found extracellularly.

### 2.3. Analysis of Metabolomic Patterns across Different Treatments over Time

To explore the metabolome-wide effects of the elicitation treatments over time, an untargeted metabolomic study was performed on cell samples, achieving the putative annotation of 1219 metabolites on day 8, 1167 on day 16, and 1007 on day 24.

The dataset for each sample was quantile normalized prior to the data analysis. As a preliminary step to uncover the patterns associated with different treatments over time, principal component analysis (PCA) with Pareto scaling was performed. Figure 3a illustrates that the metabolomic profiles of the control and COR samples are closer to each other than to the samples treated with SA at different time points. The metabolomic profiles of the control samples showed little change between days 8 and 16; but, by day 24, their profiles exhibited a considerable shift, primarily explained by Principal Component 1 (PC1), which accounts for 38.3% of the variance in the model. A similar trend was observed in the samples treated with COR, where the changes throughout the experiment were less abrupt compared to the control samples. The variance observed among the different time points in the SA treatment group was markedly distinct from the other two treatments, as indicated by PC2 (which accounts for 20.3% of the variance). This component revealed a highly significant change between the samples collected on day 8 and day 16. Additionally, it is worth noting that the profiles of the samples taken on day 16 exhibited a decrease in variance compared to those obtained on day 8, which persisted until day 24.

To enhance the differentiation between treatments, a supervised orthogonal projection of latent structures discriminant analysis (OPLS-DA) was conducted (Figure 3b). The OPLS-DA model, consisting of two predictive components and four orthogonal components, demonstrated significant statistical values: OPLS1 = 30.8% and OPLS2 = 5.25%. Moreover, the model exhibited a satisfactory explanation for the variation with an R^2^X (cum) value of 0.811, indicating a good fit. The reliability of the model was further confirmed by calculating the total variation through cross-validation, resulting in a Q^2^ (cum) value of 0.965. The analysis unequivocally revealed a substantial difference in the results among the treatments, especially between the samples treated with SA and those treated with COR. The dissimilarity between the two elicitor treatments is notable, indicating pronounced variations in the resulting metabolomic profiles. In contrast, the control conditions are positioned more closely to the treatment with COR, suggesting a certain degree of similarity or overlap in the metabolomic patterns.

### 2.4. Pathway Analysis

Annotated metabolites were subjected to statistical and FC analyses; only those meeting statistical significance (*p*-value < 0.05) and FC ≥ 2, with respect to the corresponding controls, were considered for biological interpretation at each experimental time point. The selected compounds were also uploaded to PlantCyc pathway tools [28] to assess the biosynthetic effect of the elicitors on the metabolome of *T. baccata* suspension-cultured cells at each experimental time point (Figure 4). Over time, the application of elicitors to cell cultures had a relevant impact on secondary metabolism, modulating the biosynthetic pathways of three major metabolite groups: N-containing compounds (NCCs), phenylpropanoids, and terpenoids. To a lesser extent, phytohormone biosynthesis was also modulated by the elicitors over time, thus suggesting a clear crosstalk between primary and secondary metabolism in the *Taxus* cells.

On day 8, an elicitor-dependent modulation of NCC and terpenoid metabolism was observed, revealing that SA promoted a general up-regulation whereas COR mainly exerted an inhibitory role (Figure 4a). Among the NCCs, the metabolic up-regulation attributed to SA predominantly affected the alkaloids 6,7-*O*,*O*-dimethyl-*N*-deacetylisoipecoside aglycon and (*L*)-canaline, together with the glucosinolate 6-(methylsulfanyl)-2-oxohexanoate (logFC = 10.13 for all compounds); whereas, COR was associated with the metabolic down-regulation of alkaloids, such as chelirubine and (*S*)-magnoflorine, and a wide variety of glucosinolate precursors, i.e., 2-[(5′-methylsulfanyl)pentyl]malate, 3-carboxy-10-(methylsulfanyl)-2-oxodecanoate, (*E*)-4-(methylsulfanyl)butanal oxime, *N*-hydroxypentahomomethionine, and *N*,*N*-dihydroxypentahomomethionine (logFC = −10.13 for all compounds). With respect to terpenoids, the SA elicitation drove an up-regulation of carotenoid biosynthesis, as observed for bis(β-D-glucosyl) crocetin, β-D-gentobiosyl crocetin, picrocrocin, and bixin aldehyde (logFC = 10.13 for all compounds); COR induced a down-regulation of the biosynthesis of phytoalexins and their precursors, including heliocide B1, oryzalide A, (2*E*, 6*E*)-farnesal, (6*E*)-8-hydroxygeraniol, 3β-hydroxygeraniol, and phytyl monophosphate, as well as a repression of saponin biosynthesis, such as gypsogenate and its derivatives (logFC = −10.13 for all compounds).

On day 16, the general down-regulation effect of the two elicitors was only observed in secondary metabolism, with SA having a stronger impact. In the SA-elicited cell lines, positive modulation of terpenoid biosynthesis was observed, as on day 8; whereas, the biosynthesis of phytohormones, NCCs, and phenylpropanoids was repressed (Figure 4b). The SA-driven up-regulation of steroids was reflected by an increase in metabolites, such as the taxane taxa-4(20),11-dien-5α-ol, steroid derivatives of ergostenol, and stigmastadienol; phytoalexins, such as 16-α-hydroxygypsogenate-28-β-D-glucoside, and 3β-hydroxyparthenolide; and the carotenoid norbixin (all of them showing logFC = 11.05). In turn, COR also induced a clear down-regulation of secondary metabolism at this time point, most apparently in a decrease in the biosynthesis of phytoalexins, such as pinocrocin, desoxyhemigossypol-6-methyl ether, and gypsogenate-28-β-D-glucoside (all of them with logFC = −11.05). Regarding NCCs, both elicitors had a negative metabolic effect, including on alkaloids, such as galanthamine, strictosidine aglycone, and (6*S*)-hydroxyhyoscyamine, and glucosinolates, such as (*E*)-1-(glutathion-*S*-yl)-*N*-hydroxy-ω-(methylsulfanyl)hexan-1-imine and (*E*)-1-(glutathion-*S*-yl)-*N*-hydroxy-ω-(methylsulfanyl)butan-1-imine (all of them showing logFC = −11.05 for both elicitors). A broader range of metabolites was affected by SA, resulting in a greater down-regulation effect. Finally, a similar reduction in biosynthesis was observed for phenylpropanoids, particularly phenolic compounds, such as 3″-hydroxy-geranylhydroquinone, 3,6,7-trimethylquercetagetin, and *trans*-5-*O*-(4-coumaroyl)-D-quinate (logFC = −11.05 for both elicitors). It is important to note that on day 16, SA negatively impacted the biosynthesis of phytohormones, especially *cis*-zeatin riboside, gibberellin A_25_, and (+)-*cis*-abscisate (logFC = −11.05 for all of them); whereas, COR had a slight positive effect, essentially on gibberellin biosynthesis (logFC = 2.15–11.05).

On day 24, the final experimental time point, the same general trend was observed as on day 16, with a drastic down-regulation of secondary metabolism induced by both elicitors, affecting mainly phytohormones, NCCs, and phenylpropanoids (Figure 4c). In contrast, both elicitors induced the up-regulation of terpenoid biosynthesis, with the effect being more evident for COR than SA. This elicitor-mediated induction was mostly apparent in the accumulation of biosynthetic precursors, such as (2*E*, 6*E*)-farnesoate and presqualene diphosphate, as well as a wide range of phytoalexins, including gypsogenate-28-β-D-glucoside, dehydroabietate, kauralexin B1, and the carotenoid anidorubin (all of them with logFC = 9.67 for both elicitors). The metabolism of phenylpropanoids was repressed at all levels, including simple compounds, such as benzoyl-β-D-glucopyranose, early intermediates, such as coumaroyl derivatives, and polyphenols, i.e., (2*R*, 3*S*, 4*S*)-leucodelphinidin, 3,7,3′,4′-tetramethylquercetin, and isoliquiritigenin (logFC = −9.67 for all compounds and both elicitors). With respect to NCC biosynthesis, a generalized down-regulation was observed, mostly affecting glucosinolates and their precursors, such as (*E*)-1-(glutathion-*S*-yl)-*N*-hydroxy-ω-(methylsulfanyl)hexan-1-imine and *N*,*N*-dihydroxy-tetrahomomethionine, respectively, as well as alkaloids, such as coniine and tabersonine.

Overall, the results from the metabolomic profile regarding elicited *T. baccata* cell suspension cultures suggested an elicitor- and time-dependent behaviour. Regarding terpenoid biosynthesis, SA promoted a sustained induction over time from day 8 onwards; whereas, a delayed induction by COR was observed, being effective only on day 24. COR inhibited NCC biosynthesis throughout whereas SA promoted an early up-regulation on day 8, followed by a decline. Finally, considering phenylpropanoid biosynthesis, a coordinated response was observed in the two elicitors, both promoting an early up-regulation, followed by a strong repression, as observed on days 16 and 24. Due to the importance of the taxane biosynthetic capacity of *Taxus* spp. cells, two universal precursors and their derivatives were analyzed in the elicitation experiments: L-phenylalanine and its derivatives (*N*-hydroxy-L-phenylalanine and *N*,*N*-dihydroxy-L-phenylalanine) and geranylgeranyl diphosphate (GGPP) and its derivative (dihydrogeranylgeranyl diphosphate). Furthermore, five intermediates involved in PTX biosynthesis were targeted (taxa-4,11-diene, taxa-4(20),11-dien-5α-ol, 10β,14β-dihydroxytaxa-4(20),11-dien-5α-yl acetate, taxa-4(20),11-dien-5α,13α-diol and baccatin III), as well as two PTX analogues (taxusin and 7β-hydroxytaxusin). Throughout both elicitation treatments, the precursors were strongly down-regulated, suggesting their fast consumption formed downstream compounds within the pathway. The intermediates exhibited a similar pattern, except for taxa-4(20),11-dien-5α-ol, which clearly accumulated on day 16 under SA elicitation. On the other hand, significant levels of the PTX competitor 7β-hydroxytaxusin accumulated under COR elicitation on days 8–16 (logFC = 2.18 and 1.68, respectively), its synthesis being strongly repressed thereafter (logFC = −2.22). In SA-treated samples, the biosynthesis of taxusin was strongly down-regulated for longer (logFC= −11.05 and −3.98, on days 16 and 24, respectively).

Finally, based on the previously mentioned criteria (*p*-value and FC), a metabolomic fingerprint was defined for each treatment of elicitation, focusing on the secondary metabolism in comparison to the control. To establish this fingerprint, we selected the samples corresponding to day 8, as they exhibited the highest values of total taxane production. The metabolites that define each treatment are listed in Appendix A, where positive values are characteristic of each treatment while negative values are directly associated with the control sample.

### 2.5. Transcript Profile of Taxane Biosynthetic Genes

The expression levels of the most important genes involved in the taxane biosynthetic pathway (*GGPPS*, *TXS*, *DBAT*, *BAPT*, *DBTNBT*, *T7βOH*, and *PAM* genes) and the *DXR* gene of the MEP/DOXP pathway were determined by qRT-PCR. At first sight, the results revealed that most of the target genes were more susceptible to SA than COR elicitation as their maximum gene expression was detected under those conditions. In addition, it should be noted that COR elicitation generally induced an earlier maximum gene response at 6 h to 12 h post-elicitation, compared to 24 h to 72 h after SA elicitation. In control conditions, the gene expression was extremely low, except for the *GGPPS* and *DXR* genes.

Briefly, after SA elicitation, the expression profile of the *TXS*, *DBAT*, and *BAPT* genes exhibited a constant and gradual increase until a maximum was reached at 48 h in the case of the *BAPT* gene (13.06-fold increase) or 72 h for the *TXS* and *DBAT* genes (17.03-fold and 9.45-fold increase, respectively). COR treatment induced the highest *DBTNBT* transcript levels, which were reached after 6 h of elicitation (8.06-fold increase).

On the other hand, the expression of the *T7βOH* hydroxylase gene showed a pattern of isolated peaks, reaching a maximum at 72 h post-SA-elicitation (367.98-fold increase), which was 15.25-fold higher than the maximum expression under COR elicitation (224.12-fold at 6 h). Similarly, the *PAM* gene was clearly overexpressed after the addition of SA, with the levels peaking at 24 h (39.41-fold increase) to decrease thereafter. Under COR elicitation, two clear peaks of *PAM* gene expression were detected at 12 h and 72 h (approx. 30-fold increase).

Finally, it is noteworthy that the expression levels of the *DXR* gene were only doubled by SA throughout the experiment. In contrast, COR caused the maximum gene enhancement at 72 h; although, the values were the same as in control conditions (4-fold increase). In the case of *GGPPS*, the addition of SA clearly increased its expression at 48 h and 72 h (7.15-fold and 11.28-fold increase, respectively); whereas, the values under COR elicitation were similar to those of control conditions.

### 2.6. Biomarkers

With the purpose of identifying relevant markers related to PTX production levels, an OPLS-DA analysis was conducted. As a first step, the taxane content dataset for the end-product of the biosynthetic pathway, PTX, was classified into high, medium, and low categories. This classification was achieved using quartile calculation, where the first quartile represented the group of samples with the lowest content and the third quartile contained the samples with the highest values. Only these two quartiles were used for the multivariate analysis. Once each sample was assigned to a quartile, it was linked to its corresponding metabolomics data. Subsequently, two separate OPLS-DA analyses were performed after Pareto scaling: one comparing the control and COR treatments and the other comparing the control and SA treatments. To facilitate the visualization of the discrimination models in terms of biomarkers, an S-plot [29] was utilized to filter potential compounds from the metabolomics data (see Figure 5).

The S-plot of the Control vs. COR model (Figure 5a) illustrates the magnitude (modelled covariation) and reliability (modelled correlation) of each compound. Putative biomarkers were identified based on a small set of compounds exhibiting high magnitude (≥|0.1|) and reliability (≥|0.75|). In this particular model, only positively correlated biomarkers were discovered. The compounds identified as biomarkers were as follows: 3-[(6′-methylsulfanyl) hexyl]malate, 2-(6′-methylthio) hexylmalic acid, salicyl-HCH, N6-dimethylallyladenine (2iP), benzyladenine, 6-hydroxymellein, 5-hydroxy-coniferaldehyde, and 5-methoxyfuranocoumarin (Bergapten). On the other hand, the S-plot of the Control vs. SA model (Figure 5b) exclusively reveals negatively correlated biomarkers. The compounds identified as the biomarkers in this model were kauralexin A2 and taxusin. The compounds situated in the extreme regions of the S-plot are deemed crucial for distinguishing between various PTX production groups. In the case of COR treatment, the aforementioned metabolites could be regarded as an alternative way of monitoring high PTX content in samples. Conversely, in the SA treatment, the presence of kauralexin A2 and taxusin could serve as an indicator of low production levels.

## 3. Discussion

Although *Taxus* species are the major natural source of taxanes for extraction, a large number of other secondary metabolites of industrial or pharmacological interest, such as flavonoids, lignans, and volatile components, have also been isolated and identified in this plant genus [30,31]. Due to such phytochemical heterogeneity, an untargeted metabolomics approach was performed here to systematically unravel and analyze the metabolic response attributed to the elicitation of *T. baccata* cell suspensions. Additionally, taxane production, biomass formation, and expression levels of several genes involved in taxane biosynthesis were also studied.

The growth of *T. baccata* cell cultures in an optimized two-stage system not only induced an enhancement of the taxane production [32] but also promoted cell growth throughout the experiment, as evidenced by a 17% increase relative to the inoculum under unelicited conditions. However, both elicitor treatments resulted in a decrease in biomass, especially under COR elicitation, as previously observed in other studies [33,34,35]. The more marked decrease in cell growth observed in COR-treated samples could be due to a reduced number of living cells as cell viability decreased to 72% over the study period whereas higher viability rates were observed in control and SA-treated samples (87% and 81%, respectively). Similarly, Patil et al. [36] discovered that MeJA repressed cell growth in elicited *T. cuspidata* suspension cultures after 3 days as a consequence of inhibiting cell cycle progression at the G1/S transition.

COR is a non-host-specific phytotoxin produced by *Pseudomonas syringae*, which acts as a mimic of the isoleucine-conjugated form of jasmonic acid (JA-Ile), thus activating the host’s jasmonate signalling pathway [33]. JA-Ile and its related signalling compounds, like 12-oxo-phytodienoic acid (12-OPDA) or MeJA, control a wide range of plant functions, primarily activating secondary metabolism in plant defensive responses [37]. In this regard, Koo et al. [38] and Heitz et al. [39] indicated that the more potent and rapid effect of COR on secondary metabolism is due to its chemical structure, with the rigid cis-orientation of its bicyclic skeleton providing more stability. This characteristic prevents its transformation into less active forms, such as 12OHJA (also known as tuberonic acid), by cytochrome P450 (CYP94B3), which is involved in JA-Ile turnover; thus, COR may not be susceptible to signal attenuation.

Stimulation of the JA signalling pathway through the addition of COR leads to an increase in the concentrations of JA-Ile and its related compounds, whose intracellular transduction signals may interfere with other signalling pathways, such as those of SA and auxins [40]. In the present study, the biosynthesis of less-active hydroxylated forms of JA, such as tuberonic acid [41], was repressed on day 8, after the addition of both elicitors; although, this occurred more prominently in the case of COR. Therefore, JA may activate self-defence systems in plants, including the formation of secondary metabolites, such as diterpenoid phytoalexins kauralexins and oryzalexins, whose biosynthesis was mainly up-regulated on day 16 of culturing. On the other hand, the biosynthesis of cytokinins and derivatives (such as cis-zeatin, cis-zeatin riboside, and N(6)-dimethylallyladenine) was found to be stimulated by both elicitation treatments on day 8 but repressed afterwards, especially under SA. Tamogami and Kodama [42] observed that exogenously applied COR induced the production of the phytoalexin sakuranetin in *Oryza sativa* leaves; whereas, in their previous studies, kinetin and zeatin were found to significantly counteract JA-inducible sakuranetin production at very low concentrations

Elicitation is responsible not only for the stress-driven induction of reactive oxygen species and reactive nitrogen species, affecting cell homeostasis [43], but also for the enhancement and accumulation of bioactive secondary metabolites via an overexpression of biosynthetic genes. As described, the following families of metabolites were distinguished by their response to the two elicitors: N-containing secondary metabolites (predominantly alkaloids), terpenes, phenylpropanoids, and phytoalexins.

Within the phenylpropanoids, phenolic compounds, including flavonoids (flavanols, flavonols, and flavones), lignans, stilbenes, and phenolic acids, accounted for the differential annotation between the elicitors. In general, phenylpropanoid biosynthesis in *T. baccata* cell suspensions was up-regulated during the early stages of elicitation by both treatments, followed by a strong general down-regulation. However, at the end of the experiment, SA and COR up-regulated the biosynthesis of certain anthocyanins and flavanones. Additionally, COR independently boosted the biosynthesis of coumarins. The biosynthesis of coumarin-related compounds has also been enhanced by the exogenous application of MeJA in other plant species, including the diploid and tetraploid plants of *Matricaria chamomilla* [44] and the hairy root cultures of *Pelargonium sidoides* [45]. Regarding alkaloid biosynthesis, the main groups found in *Taxus* cell suspensions were indole, tropane, quinolizine, and polyketide-derived alkaloids. These compounds experienced a general down-regulation due to the elicitation treatments in the medium and long term.

The terpenoid biosynthetic pathway is the source of the most important secondary metabolites in *Taxus* spp.: monoterpenoids, sesquiterpenoids, diterpenoids (mainly PTX and related taxanes), triterpenoids, and carotenoids. Although elicitation with COR and, to a greater extent, SA significantly enhanced PTX production, the total taxane content found in this study was low, suggesting that this *T. baccata* cell line was naturally a low taxane-producer and, thus, not suitable for the biotechnological production of PTX, as reported in other studies [46,47,48,49]. Regarding the low taxane yield, it should be noted that the response of the taxane biotechnological production system to elicitation is highly variable, depending on multiple factors. These factors include those related to the elicitor itself, such as its type, concentration, or duration of treatment. Additionally, the response is influenced by the species used and the specific developmental stage of the culture during the experiments [50].

The transcriptional profile study showed that SA and, to a lesser extent, COR induced clear overexpression, not only of the genes directly involved in taxane biosynthesis but also of the *GGPPS* gene, encoding a key enzyme involved in the formation of terpenoid precursors. In relation to the *GGPPS* gene, Banmeshin et al. [51] also reported its positive regulatory role in enhancing taxane production under light elicitation. However, it did not limit the rate of taxane biosynthesis. The authors also demonstrated that the transcriptional expression of the *DXS* gene (deoxy-D-xylulose-5-phosphate synthase) did not increase under light conditions, similarly to the control conditions. The present study observed a similar situation regarding the *DXR* gene, under both elicited and control treatments. A possible explanation for the low taxane contents detected may be that PTX biosynthesis had to compete with the metabolism of other terpenoid subclasses [52]. The higher availability of precursor pools and the potential deviation of the carbon flow to the isoprenoid pathway may explain the up-regulation observed in the biosynthesis of carotenoids, triterpenoids, sesquiterpenoids, and monoterpenoids during the early stages of the experiment, from day 8 to day 16 in COR-treated samples and on day 16 under SA elicitation.

On the other hand, while the biosynthesis of carotenoids and triterpenoids was still being stimulated at the end of the experiment, a clear repression of monoterpenoid and sesquiterpenoid biosynthesis was observed, which was concomitant with an increase in diterpenoid biosynthesis. This finding is supported by the detected stimulation of the precursors or derivatives of these subclasses, such as farnesal for sesquiterpenoids, phytol (a hydrogenated form of phytoene) for carotenoids, and presqualene diphosphate in the case of triterpenoids and sterols.

In general, a metabolomic fingerprint obtained through untargeted analysis is suitable for discriminating between elicitation treatments and identifying characteristic biomarkers. However, the generic nature of the metabolite extraction method often leads to the obscuration or presence of metabolites of interest, such as taxanes, at concentrations below the detection limits. This limitation was encountered during our metabolomic analysis. Therefore, in this study, a specifically designed extraction method was employed to enhance the enrichment of the taxanes of interest, ensuring their accurate quantification.

Finally, the use of relevant biomarkers, whose presence or absence is closely related to different PTX production levels, can greatly facilitate the rapid screening and subsequent selection of different *Taxus* cell lines for biotechnological purposes. The results indicated that in the SA treatment, the presence of two diterpenes, kauralexin A2 and taxusin, could serve as indicators of low PTX production levels. In effect, the biosynthesis of the PTX competitor taxusin and its analogues, such as 7β-hydroxytaxusin, were significantly increased by COR and strongly repressed by SA. These compounds are produced in the intermediate oxygenation steps and are considered dead-end metabolites in PTX biosynthesis [53]. Thus, an induction of their biosynthesis may consume the limited intermediates available and block the metabolic flow towards PTX formation. In the case of COR treatment, the detection of high levels of certain compounds, including coumarins, like 5-methoxypsoralen (bergapten) and the dihydroisocoumarin 6-hydroxymellein (6-HM), or cytokinins, such as benzyladenine and N6-dimethylallyladenine (2iP), may be indicative of highly PTX-productive cell lines.

## 4. Materials and Methods

### 4.1. Plant Material

Cell suspension cultures were established from a stable callus line of *T. baccata* derived from sterilized young stems, as previously described by Cusido et al. [54]. Sterile stems were placed in contact with the induction medium, which consisted of Gamborg’s B5 medium [55] supplemented with 2× B5 vitamins, sugars, and hormones, as described by Exposito et al. [16]. After 3–4 weeks, the different callus tissues formed were separated from the explants and placed together. Calli were grown in solid Gamborg’s B5 growth medium (GM) supplemented with 2× B5 vitamins, sugars, and hormones, as described by Exposito et al. [16], and the callus size was between 2 and 3 cm in diameter. The cells were cultured in the GM at 25 °C in darkness and subcultured every 2 weeks to obtain enough friable and vigorous calli to establish cell suspension cultures. Regarding the period spanning from the induction to the date of the experiments, the taxus cells went from 0 to about 7 months old.

### 4.2. Cell Suspension Culture Conditions and Elicitation

Cell suspensions were cultured using a two-stage system, as described by Cusido et al. [54] and Palazon et al. [32]. After 14 days of culturing, taxane production was stimulated by adding the elicitors at the beginning of the second phase of culturing [33]. Elicitation was performed with 1 µM of coronatine (COR) and 150 µM of salicylic acid (SA) (Sigma Aldrich, St. Louis, MO, USA), which were previously filter-sterilized and added to the cell suspensions. The SA concentration was selected based on Loc et al. [56] and Alcalde et al. [27] studies.

### 4.3. Biomass Determination and Viability Assay

Fresh weight (FW) was determined by filtering the cells with 80 µm Nylon filters. The cells were freeze-dried to obtain the dry weight (DW) and perform both the taxane extraction and the untargeted metabolomics analysis. Cell viability was evaluated as described by Exposito et al. [16]. Samples were harvested on days 0, 8, 16, and 24 of the treatments.

### 4.4. Metabolite Extraction for Untargeted Metabolomics Analysis

Lyophilized cell samples were extracted, as previously reported by Garcia-Perez et al. [57], using the solvent mixture of MeOH/H_2_O/HCOOH (80:19.9:0.1 *v*/*v*/*v*) at a concentration of 50 mg/mL; this mixture was vortexed for 2 min until homogenization and then sonicated for 10 min at 25 °C. Afterwards, samples were centrifuged at 12,000× *g* for 10 min at 4 °C (Eppendorf 5810R, Hamburg, Germany). Finally, the supernatants were collected and filtered (0.22 µm PVDF filters, Millipore, Billerica, MA, USA) and the extracts were either transferred into vials for analysis or stored at −20 °C until use.

### 4.5. UHPLC/Q-TOF—MS for Untargeted Metabolomics

Plant metabolites from both control and elicited cultures were screened by an untargeted metabolomics approach using a tandem system based on a quadrupole-time-of-flight (QTOF) mass spectrometer coupled with an ultra-HPLC chromatographic system (UHPLC/Q-TOF—MS), as previously described by Lucini et al. [58]. The injection volume was 12 µL and the process was repeated twice for each sample (*n* = 6). The mass spectrometer was set in positive polarity and SCAN mode, selecting the extended dynamic range mode of 100–1200 *m*/*z*. The annotation of all the compounds found was performed following their monoisotopic precise mass and isotopic pattern and was expressed as mean values of the overall identification score using MassHunter Profinder v. 10.0 software (Agilent Technologies, Santa Clara, CA, USA) and the database imported from PlantCyc v. 15.0.1 (Plant Metabolic Network; freely available on http://www.plantcyc.org; accessed on 20 October 2022). Finally, identification was carried out according to Level 2 (putatively annotated compounds), as displayed by the COSMOS Metabolomics Standards Initiative [59].

### 4.6. Statistical Analysis of the Untargeted Metabolomics Dataset

Metabolomics-obtained raw data were statistically analyzed and interpreted using the bioinformatic software Mass Profiler Professional v. 12.6 (Agilent Technologies, Santa Clara, CA, USA). Moreover, a supervised modelling by orthogonal projection to latent structure discriminant analysis (OPLS-DA) was carried out using SIMCA v. 16.0.2 software (Umetrics, Sweden) to discern the effects of the different elicitation treatments. The model parameters explaining the goodness-of-fit (R^2^Y) and the goodness-of-prediction (Q^2^Y) were also recorded. Finally, a multifactorial ANOVA analysis was performed to select the compounds exhibiting a significantly different abundance between treatments (*p*-value of <0.05), followed by Tukey’s post hoc test and Bonferroni correction. A fold change (FC) analysis was also carried out to reveal featured variations in the abundance of compounds (FC ≥ 2). Thus, only those compounds meeting both criteria (*p*-value and FC) were uploaded into the Omic Viewer Pathway Tool of PlantCyc software (Stanford, CA, USA) for biochemical interpretations [60].

### 4.7. Total Taxane Extraction and Quantification

Taxanes were extracted from the lyophilized biomass, as was conducted in Perez-Matas et al. [35]. Finally, all samples were resuspended in 500 µL of MeOH and were filtered (0.45 µm PVDF filters, Millipore, Billerica, MA, USA) prior to analysis. HPLC analyses were performed, as was conducted in Sabater-Jara et al. [17]. Taxanes were quantified by integrating the corresponding peak of each target compound in the standard calibration curve: 10-deacetylbaccatin III (DABIII), baccatin III (BACIII), cephalomannine (CEPH), and paclitaxel (PTX). All standards were obtained from Abcam (Cambridge, UK).

### 4.8. Transcript Analysis

Total RNA was isolated from 200 mg of frozen cells at 0, 6, 12, 24, 48, and 72 h using the Real Plant RNA Kit (REAL, Valencia, España), according to the manufacturer’s instructions. The RNA concentration of each sample was determined using a NanoDrop ND-1000 spectrophotometer (NanoDrop Technologies, Wilmington, NC, USA). Additionally, cDNA was prepared from 1 μg of RNA with SuperScript IV Reverse Transcriptase (Invitrogen, Waltham, MA, USA).

The gene expression levels were determined by qRT-PCR using SYBR Green Mastermix (Biorad, Hercules, CA, USA) in a 384-well platform system (LightCycler^®^ 480 Instrument, Roche, Basel, Switzerland). The target genes were *TXS*, taxadiene synthase; *DBAT*, 10-deacetylbaccatin III 10-O-acetyltransferase; *BAPT*, baccatin III-3-amino-13-phenylpropanoyltransferase; *DBTNBT*, 3′-N-debenzoyltaxol N-benzoyltransferase; *T7βOH*, 17-β-hydroxylase; *GGPPS*, geranylgeranyl diphosphate synthase; *DXR*, 1-deoxy-D-xylulose-5-phosphate reductoisomerase, and *PAM*, phenylalanine aminomutase. Gene-specific primer sequences were obtained from the previous studies of our research group [17,61,62] (Appendix A). The reaction mixture, primer amplification efficiency, and thermos-cycling program were handled as described previously [35]. The *TBC41* gene was selected as a reference to normalize gene expression [17,48,63]. Data were analysed using LightCycler^®^ analysis software v4.1.

### 4.9. Statistics

All data regarding biomass and viability determination, taxane quantification, and transcription profiling were expressed as the average of six independent determinations ± standard deviation. The statistical analysis was performed using Excel and RStudio software. The multifactorial ANOVA analysis followed by Tukey’s post hoc multiple comparison tests were used for statistical comparisons. A *p*-value of < 0.05 was assumed for significant differences.

## 5. Conclusions

In the present study, the application of untargeted metabolomics allowed us to examine, in depth, the differential metabolic variations in *T. baccata* cell suspensions under abiotic stress caused by the elicitors SA and COR; we could also discover how this elicitation affected gene expression and biomass formation. By identifying these potential biomarkers, a deeper understanding of the underlying mechanisms influencing taxane production levels has been gained. The results obtained contribute new data about the influence of different elicitors on the interplay between different metabolic pathways, providing new insights into the relationship between the primary and specialized metabolisms in elicited cell suspensions, as well as improving the biotechnological production of paclitaxel. These findings can have significant implications in the fields of biotechnology and metabolic engineering, contributing valuable information for the design and optimization of taxane biosynthetic pathways for industrial or therapeutic purposes.

## Figures and Tables

**Figure 1 plants-12-02696-f001:**
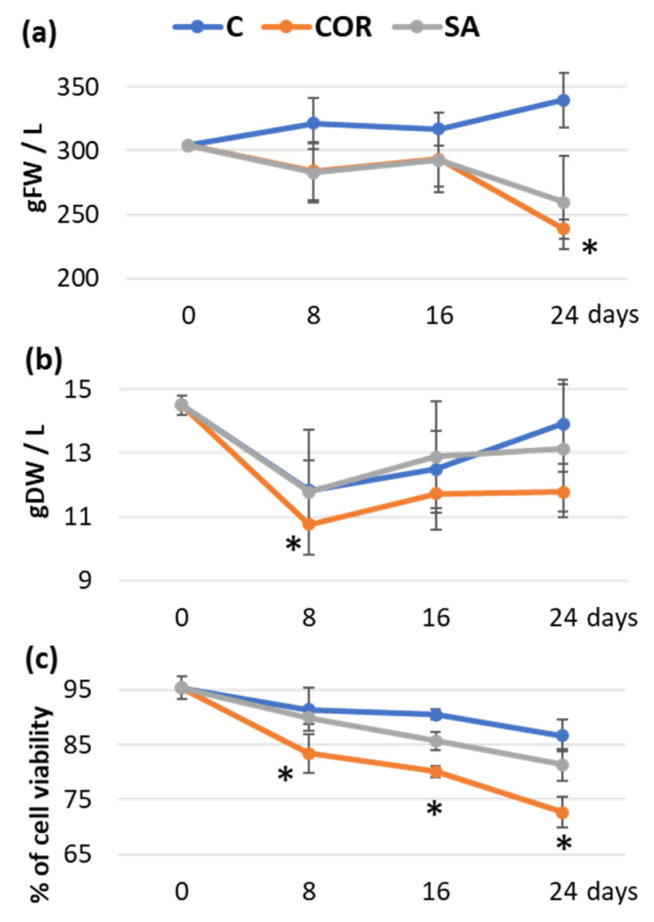
Time courses of biomass production and viability of *T. baccata* cell suspensions cultured for 24 days in the production medium in control conditions, or with the addition of 1 µM COR (coronatine) or 150 µM SA (salicylic acid). (**a**) FW in g/L. (**b**) DW expressed in g/L. (**c**) Percentage of cell viability. In all cases, the initial inoculum was of 300 g/L of fresh cells. Data represent average values from three separate experiments ± SD. Values marked by an asterisk (*****) are significantly different (*p* ≤ 0.05) according to Tukey’s honestly significant difference test.

**Figure 2 plants-12-02696-f002:**
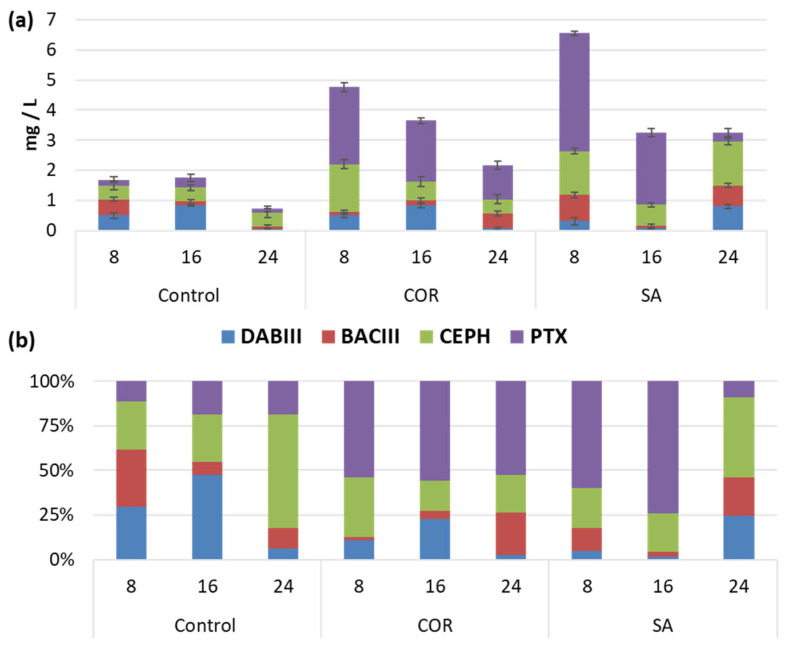
Taxane content under different treatments. (**a**) Total taxanes (expressed as mg/L) produced in *Taxus baccata* cell cultures throughout 24 days of culture in control conditions, with the addition of coronatine (COR) or salicylic acid (SA); the values are means ± SD (*n* = 6). (**b**) Taxane distribution is expressed as a percentage of individual taxane content.

**Figure 3 plants-12-02696-f003:**
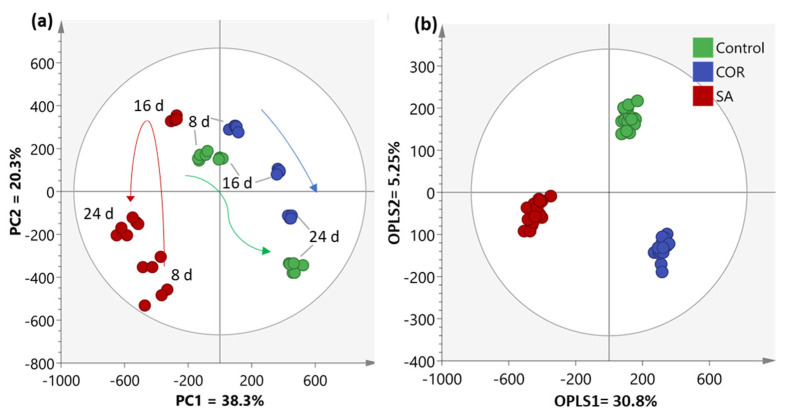
(**a**) Principal component analysis (PCA) score plot illustrating the temporal tendencies of each treatment. Each arrow represents a specific behaviour of the samples. (**b**) Orthogonal partial least squares discriminant analysis (OPLS-DA) score plot depicting the differentiation between treatments. OPLS1 shows variation related to the classes or treatments covered by the model while OPLS2 represents the remaining variation orthogonal to the predictive component.

**Figure 4 plants-12-02696-f004:**
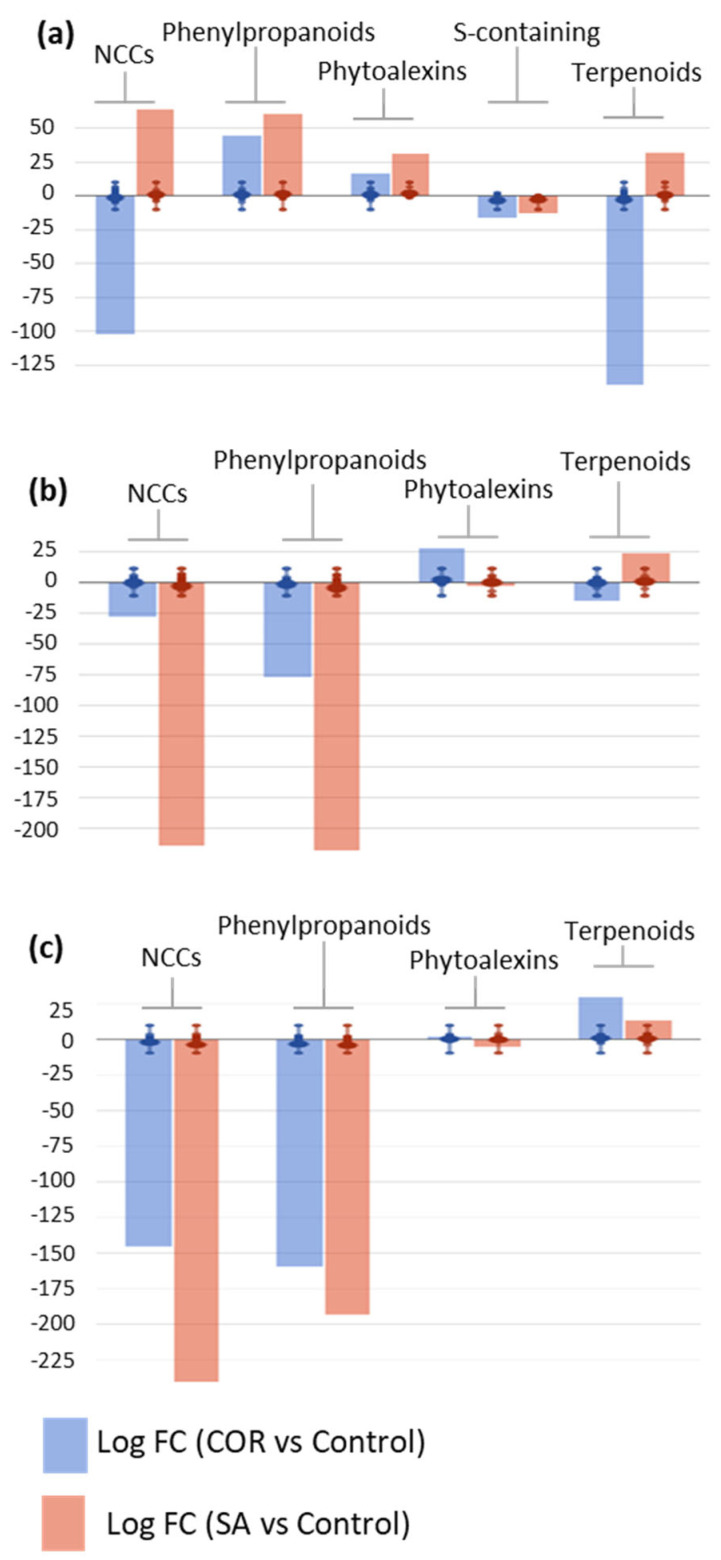
Plots of secondary metabolite biosynthesis in *Taxus baccata* cell suspensions maintained in the taxane production medium and elicited with COR 1 µM or SA 150 µM on days 8 (**a**), 16 (**b**), and 24 (**c**) of culturing. The large dots represent the average (mean) of all log fold changes (FCs) for metabolites and the small dots represent the individual log FCs for each metabolite within each family. NCCs: nitrogen-containing secondary compounds; phenylpropanoids and derivatives; S-containing: sulfur-containing secondary compounds.

**Figure 5 plants-12-02696-f005:**
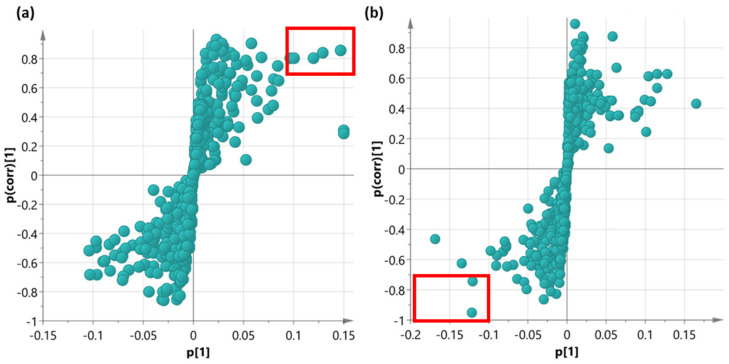
The S-plot illustrates the relationship between the magnitude of change (p[1]) and the reliability of the measure (p(corr)[1]) for each variable (compound) in the model. (**a**) This plot was obtained from the Control vs. COR OPLS-DA model. (**b**) This plot was obtained from the Control vs. SA OPLS-DA model. The red rectangle highlights compounds that meet the selected magnitude (≥|0.1|) and reliability (≥|0.75|) criteria. Green solid circles represent the compounds.

## Data Availability

Data is contained within the article and Appendix A.

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
