# Peer review of "Exploring the Interplay between Metabolic Pathways and Taxane Production in Elicited Taxus baccata Cell Suspensions"

_plants, 2023, doi:10.3390/plants12142696_

Round 1

Reviewer 1 Report

The experimental designing is good, and lot of work done. However, I do not see any novelty on this study. All these experiments are previously done by other authors either with the plant or with other plants like Andrographis.....Also patent but not one of the elicitors. The authors demand in Line 84-91, the novelty of the aim, which is biased, So, many interesting facts are already reported in their published review -Biotic Elicitors in Adventitious and Hairy Root Cultures: A Review from 2010 to 2022 (ub.edu)

Moreover, interesting papers, I came across but not cited by authors-

Future Pharmacology | Free Full-Text | Increased Production of Taxoids in Suspension Cultures of Taxus globosa after Elicitation (mdpi.com)

Gene expression pattern and taxane biosynthesis in a cell suspension culture of Taxus baccata L. subjected to light and a phenylalanine ammonia lyase (PAL) inhibitor - ScienceDirect

Taxus Cell Cultures, an Effective Biotechnological Tool to Enhance and Gain New Biosynthetic Insights into Taxane Production | SpringerLink

A common pharmacophore for taxol and the epothilones based on the biological activity of a taxane molecule lacking a C-13 side chain — Albert Einstein College of Medicine (elsevierpure.com)

patent

EP0960944A1 - Enhanced production of taxol and taxanes by cell cultures of taxus species - Google Patents 

Minor comments:

Line 16, Taxus instead of "T."

Line 48, 8 60-year-old trees?

Line-75: "LC-IT-TOF-MS", IT is not necessary

LIne 94- "GM"? also later case..

Line 96- PM?

.lot of discrimens in sentences, grammar error

Can be improved..

Reviewer 2 Report

Comments
The manuscript "Exploring the interplay between metabolic pathways and tax-2 ane production from elicited Taxus baccata cell suspensions " represents good and attractive work with reproductive statistical analysis methods. I am highly recommending this manuscript for publication with slight modifications as follows:

1. Line 503: The authors need to mention the company where they obtained the callus cell line.

2. Line 504-512: The authors need to include the size and age of the callus which has been used.

3. The language or sentence formations need to be addressed carefully for better understanding.

4. The authors used many old references for examples 2, 3, and 9.

Reviewer 3 Report

The work presents an interesting, multistep study of the metabolic response attributed to the elicitation of Taxus baccata cell suspensions, as well as taxane production, biomass formation, and expression levels of several genes involved in taxane biosynthesis. As the search for new solutions for cell suspensions is essential for the development of rich sources of anticancer drugs, the topic of the research is valid. Not only the influence of different elicitors should be studied but also the wider understanding of metabolic pathways for metabolic engineering is important. The obtained results do not give the complete answer and further research is required, but as giving new insight in comparison to so far published research, this paper would be interesting not only as the basis for further analysis of the Authors but also for other research groups. 

Generally, the study is well designed and interpreted, can arouse public interest, and can consider for publication. I read the manuscript thoroughly and I can not find any serious allegations therefore I would recommend it for publication in Plants journal.  

Author Response

Thank you for your comments.

Reviewer 4 Report

The comments are listed below:

1. How is the concentration of SA decided?

2. The fingerprint of T. baccata should be established and recognized the taxanes in fingerprint. 
